# Energy Drink Consumption, Depression, and Salutogenic Sense of Coherence Among Adolescents and Young Adults

**DOI:** 10.3390/ijerph17041290

**Published:** 2020-02-17

**Authors:** Ákos Tóth, Rita Soós, Etelka Szovák, Noemi M. Najbauer, Dalma Tényi, Györgyi Csábí, Márta Wilhelm

**Affiliations:** 1Institute of Sport Sciences and Physical Education, Faculty of Science, University of Pécs, H-7624 Pécs, Hungary; mwilhelm@gamma.ttk.pte.hu; 2Doctoral School of Health Sciences, Faculty of Health Sciences, University of Pécs, H-7621 Pécs, Hungary; soosrita8@gmail.com (R.S.); ethyfitwell@gmail.com (E.S.); 3Institute of English Studies, University of Pécs, H-7624 Pécs, Hungary; noemi_najbauer@yahoo.com; 4Department of Neurology, Faculty of Medicine, University of Pécs, H-7623 Pécs, Hungary; tenyidalma@gmail.com; 5Department of Pediatrics, Faculty of Medicine, University of Pécs, H-7623 Pécs, Hungary; csabi.gyorgyi@pte.hu

**Keywords:** energy drinks, adolescents, young adults, sense of coherence, depression

## Abstract

The prevalence of energy drink consumption has increased in Hungary over the past 10–15 years. This study assesses the frequency, motivations, and adverse effects of energy drink consumption, and examines how the process of becoming a regular consumer is connected with sense of coherence and depression symptoms. A total of 631 high school and college students were assessed using the Depression Scale (BDS-13) and Sense of Coherence Scale (SOC-13). Logistic regression models were fit to test the effect of and links between factors influencing addiction to energy drink use. A total of 31.1% (95% CI: 27.4–34.7) of those surveyed consumed energy drinks, 24.0% of those affected consumed the energy drink with alcohol, 71.4% (95% CI: 64.7–77.3) experienced adverse effects following energy drink consumption, and 10.2% (95% CI: 6.7–15.2) experienced at least four symptoms simultaneously. The most common motivations of consumption were fatigue, taste, and fun. Obtained by multivariate logistic regression models, after adjustment for sex and age, SOC and tendency to depression had a significant influence on the respondents’ odds of addiction. The tendency to depression increases the chances of addiction, while a strong sense of coherence diminishes the effects of depression. Young people in Hungary have been shown to consume energy drinks in quantities that are detrimental to their health. Complex preventive measures and programs are needed to address the problem. Families and educators should strive to instill a strong sense of coherence in children from an early age.

## 1. Introduction

Energy drinks (EDs) are a group of commercially available beverages that are usually carbonated, contain caffeine [1], taurine, glucuronolactone, carbohydrates, and vitamins with various dissolution properties [2,3,4,5], niacin, pyridoxine, riboflavin (B2), ginseng extracts, inositol (B8), guarana (caffeine, theobromine, and theophylline), gingko biloba, medicinal herbs, and L-carnitine [1,6]. The favorable effects of EDs depend on the ingredient combinations [7,8].

A representative survey in Australia has shown that elementary and high school populations consume significant amounts of caffeinated drinks including EDs [9]. In the UK, the rate of consumption of EDs grew by 155% between 2006 and 2014. Young people in the UK consume more EDs (3.1 per month) than their continental counterparts (2.1 per month) [10]. In the U.S., energy drinks are the second most common dietary supplements used by young people [1]; a large sample demonstrated that 30% of high school students admitted to being regular consumers [11], and there is a strong correlation between ED consumption and smoking, alcohol consumption, and drug abuse [12]. Numerous studies have pointed out the adverse health effects of EDs and their connection to destructive behaviors [10]. ED consumption is widespread among American teenagers (13–17 years), and the rate depends on demographic, psychosocial, lifestyle, and substance abuse factors [13]. There is a strong correlation between the consumption of EDs and increased soft drug use, which in turn is linked to increasing drug abuse in general [14]. Surveys of American high schoolers indicate a strong link between ED consumption and hyperactivity or general lack of attentiveness [15]. A study of 15–16 year-olds demonstrated a strong correlation between caffeine consumption, aggressive behavior, and behavioral disorders [16]. 

The safe limits of caffeine consumption are yet to be determined, but data suggest that healthy adults may consume up to 400 mgs a day without adverse effects to their health [17]. Caffeine is a major ingredient in EDs, and excessive consumption may result in acute toxicity, with symptoms like tachycardia, vomiting, arrhythmia, seizures, and death [18]. 

Researchers report on the adverse effects of ED consumption [1,19]. Huhtinen and colleagues showed in a Finnish sample (12–18 years) that there was a strong correlation between daily ED consumption and symptoms such as headaches, sleep disturbances, and fatigue. Known side effects of excessive caffeine consumption include tachycardia, tremors, high blood pressure, and in the most serious cases, sudden death [20,21]. Among Icelandic children (10–12 years), results have shown that stomach pains and headaches as well as insomnia were more common among ED consumers [22]. ED use may also be accompanied by seizures, anxiety, nervousness, hallucinations, migraines, gastrointestinal disease, metabolic acidosis, insomnia, arrhythmia, chest pain, and other cardiovascular complications [23,24].

Mixing EDs with alcohol is popular among teenagers and college students [25], bringing further high-risk behaviors such as excessive alcohol consumption, smoking, and drug abuse [26]. A significant correlation has been found between mixing alcohol with EDs and smoking, alcohol, and cannabis consumption [27]. Those who drank alcohol with Eds were also more likely to use marijuana, ecstasy, and cocaine [28], and to become alcoholics [29].

The prevalence of energy drink consumption has increased in Hungary over the past 10–15 years. According to a European survey covering 16 member states including Romania and Hungary, 30% of adults (18–68 years) and 68% of adolescents (10–18 years) interviewed were energy drink consumers [30]. “In Hungary and all around the world, the incidence of consumption of energy drinks together with alcohol has increased among adolescents and young adults.” [31] In a quantitative study in Hungary, 1459 persons were examined. The target-group consisted of the 10–26 year-old population. According to the collected data, 81.8% of the participants had consumed at least one energy drink and 63.3% had tried several products of the same kind [32].

The aim of the present study was to assess the prevalence of ED consumption between Hungarian high school and college students and analyze their motivations as well as the effects and side effects of ED use. We looked for a correlation between the sense of coherence of young people, depression, and ED consumption. Based on these, we aimed to estimate the relative significance of the factors contributing to ED addiction. 

## 2. Methods

### 2.1. Participants and Procedure

In March and April 2017, 652 young people chosen randomly from a high-school and college-age population in Hungary’s Southern Transdanubian region filled out anonymous pencil and paper questionnaires. A total of 96.8% (631) of the questionnaires were valid (284 male and 347 female respondents). 

The questionnaire requested sociodemographic information (sex, age, education level), queried ED use, and other mood enhancers and stimulants as well as exercise habits, and made use of two tests validated and accepted in Hungary: Antonovsky’s Sense of Coherence (SOC-13) and the 13-Item Beck Depression Inventory (BDI-13).

### 2.2. Measurements

Participants were asked about their ED consumption, and if the answer was yes, amounts were also asked. The “small can” (200 mL) was set as the unit of measure. Those consuming EDs from a bottle were asked to indicate the bottle size (500 mL), which was converted to “small can” units. Describing the frequency of consumption, the following categories were introduced: no consumption; very rare consumption; 1–2 small cans a month; 1–2 small cans a week; more than 2 small cans a week; 1 small can a day; and more than one small can a day. We considered those who consumed EDs very rarely to be virtual non-consumers. Those who consumed EDs on a weekly basis were considered regular consumers, while those using EDs on a daily basis were considered as “addicted”. 

An important factor was whether the participant consumed EDs alone or mixed with alcohol or other caffeinated drinks. Participants were asked about the consumption of other mood enhancers or stimulants and were requested to name these. Based on significant influential factors, categories were grouped together in order to estimate the odds of addiction. Non-consumers and those with very low rates of consumption were compared with those who used EDs regularly (i.e., consumed 200–400 mL several times a week). 

Examining the short-term effects of ED consumption, questions were asked: “Following the consumption of an ED, do you experience any of these symptoms? Feel free to mark as many as apply.” Twelve possible symptoms (i.e., headache, nausea, weakness, tremors, dizziness, loss of consciousness, insomnia, irritability, tachycardia, breathlessness, fear, diarrhea) were listed, next to which participants indicated their experiences by circling “yes” or “no”.

To understand the motivations of ED consumption, the following question was asked, “If you do consume EDs, why?” and possible motivations were given to choose from (mark several motivations when applicable): for fun, it tastes good, it stimulates me, against fatigue, it revs me up, to quench my thirst, it’s a cool/trendy thing to do, to enhance performance, before working out, other reasons, namely:……

We assumed other family members and friends might influence ED consumption, so questions were asked also regarding these factors.

### 2.3. Sense of Coherence

SOC was measured using an abbreviated 13-item adaptation of the original Orientation to Life questionnaire of Antonovsky [33]; the validated Hungarian version was used in a survey of 16–17 year-olds [34]. We used a 7-point rating scale (mostly ranging from 1 = very rarely to 7 = very often) to measure regularity and feelings that were experienced, as described by the particular item. “It often happens” indicated a weaker sense of coherence, while a higher total score would correspond to a stronger sense of coherence. The scale includes four items referring to meaningfulness: (e.g., “Do you have the feeling that you don’t really care about what goes on around you?”), five comprehensibility items (e.g., “Has it happened in the past that you were surprised by the behavior of people you thought you knew well?”), and four manageability items (e.g., “Has it happened that people you counted on disappointed you?”). Employing Antonovsky’s holistic idea of the SOC scale, we applied the common practice of using total scores (suggestions based on the sum of the scores attained in response to the given items). In this sample, Cronbach’s alpha for the scale was 0.80. In logistic regression models, however (to aid comprehensibility), a weak sense of coherence was matched with a higher score because a weaker sense of coherence is a significant risk factor in estimating the chances of addiction.

### 2.4. Beck Depression Inventory 

To understand the motivations for and the effects of ED consumption, the modified version of the Beck Depression Inventory [35] (see Appendix A) was used (the Hungarian version of the Beck Depression Inventory, Cronbach’s alpha of the sample was 0.88). 

The Inventory utilized consisted of 13 items, with four response options per item, presented on a scale of 0 to 3. For example, measuring pessimism (Item 2), the response options ranged from: “I am not particularly discouraged about the future” (score 0) to “The future is hopeless and things cannot improve” (score 3). 

### 2.5. Statistical Analysis

Statistical analysis was carried out with SPSS 25.0 for Windows (IMB SPSS Inc., Chicago IL, USA). First, Pearson’s chi-square test was used, then the Mann–Whitney test, to assess the role of the sex and age of participants. The levels of significance for all cases were determined at 5%.

Logistic regression models were fit to test the connections between ED consumption, sense of coherence, and depression. A common factor analysis (PCA) was performed for the items belonging to sense of coherence and depression to assess their respective roles (independent of one another) in the development of addiction. 

### 2.6. Ethics

The study protocols were in accordance with the latest version of the Declaration of Helsinki. The Institutional Review Board of the University of Pecs–Clinical Center, Regional, and Institutional Research Ethical Committee approved the study (reference number 6456). All participants and (if the participants were under 18 years of age) their parents were informed about the study and all of them provided informed consent.

## 3. Results

### 3.1. Sample Characteristics

Sample characteristics are presented in Table 1. A total of 286 (45.3%) of the respondents were high schoolers, while 345 (54.7%) were college students. The ratio of the sexes did not vary significantly between the two samples (*p* = 0.358). Only 19.7% of respondents did not use EDs at all (had never tasted an ED in their life) and 31.0% of respondents consumed EDs at least once or twice a month. 

Rates of ED consumption differed between males and females, and the distribution of values also showed significant (*p* < 0.001) differences, where males consumed EDs more frequently than females. 

### 3.2. Motivations for Energy Drink Consumption

Concerning the motivations of ED use, most respondents marked fatigue as their primary choice, followed by taste. The order of motivations in the case of male and female participants did not differ significantly, but males were more likely to use EDs for fun or before working out, while females tended to consume EDs to fight fatigue. (Table 2.)

Respondents were much more likely to consume EDs if their parents, siblings, or friends were also consumers. In all three cases, the chi-square test yielded a *p* < 0.001. 

From the point of addiction, daily consumers marked taste and thirst quenching as motivational factors for ED use significantly more often than those who consumed the drinks less frequently. For male regular consumers marking taste as their primary motivation, OR was 2.6 (95% CI: 1.4–4.6), while for females OR was 2.2 (95% CI: 1.2–4.1). Regular consumers marked thirst quenching as their primary motivation: for males, OR = 2.8 (95% CI: 1.1–7.2); for females, OR = 9.9 (95% CI: 2.9–20.9).

### 3.3. Energy Drink Consumption and Simultaneous Use with Alcohol

A total of 24% (95% CI: 17.9–30.0 %) of ED consumers drank it alongside or mixed with alcohol, and 21.2% (95% CI: 18.0–24.4 %) of respondents also used other stimulants. Among ED consumers, there is a higher rate of stimulant use than among non-consumers where 26.0% (95% CI: 19.8–32.2 %) (*p* = 0.049) reported using other stimulants; 11.9% (95% CI: 9.4–14.4 %) of respondents reported drinking coffee, but there was no correlation between coffee consumption and ED use (*p* = 0.651).

### 3.4. Side Effects of Energy Drink Consumption

A total of 71.4% (95% CI: 64.7–77.3 %) of respondents reported experiencing adverse ED effects, and 10.2% (95% CI: 6.7–15.2%) simultaneously experienced four or more of the symptoms on our list. There was no significant difference between male and female respondents in the symptoms experienced. In both sexes, the primary side effects were tachycardia, insomnia, and tremors (Table 3).

Examining the frequency of individual side effects, the chi-square test and Fisher’s exact test indicated a significant difference between male and female respondents only in one side effect: fear (*p* = 0.013). 

Considering ED consumption alongside alcohol or other stimulants, the frequency of the various side effects showed a very strong association (results of the chi-square test) with symptoms such as nausea (*p* = 0.020), irritability (*p* = 0.001), and fear (*p* = 0.010). The number of side effects experienced was directly proportional to the frequency of ED consumption (Spearman correlation coefficient was 0.496; *p* < 0.001). 

### 3.5. Multivariate Analysis of Factors Leading to Energy Drink (ED) Addiction 1

We examined the relative roles of variables linked to ED consumption to determine the order of variables playing a significant role. The dependent variable was: daily vs. less frequent ED consumption. First, the predictor variable was only BDI-13, while in the next round, we also added SOC-13 scores as a predictor variable (Table 4).

Results obtained via multiple independent variable models confirmed that sex, age, depression, and sense of coherence significantly contributed to ED consumption and increased the frequency of consumption among those who were already consumers (males, somewhat older, and those with a weak sense of coherence consumed EDs more frequently).

College students in the sample were on average three years older than their high school counterparts; however, substituting the category of age with school type (high school vs. college), similar results were obtained. 

### 3.6. Multivariate Analysis of Variables Leading to ED Addiction 2

In model 1, depression was a significant factor. In model 3, however, the depression and sense of coherence factors were entered together, and the significance of a tendency to depression disappeared. This might be explained by the fact that there is a strong correlation (Spearman’s rho = 0.531, *p* < 0.001) between the tendency to depression and a lower sense of coherence (SOC-13_rev). 

Analyzing the “active ingredients” of depression and sense of coherence, which are quasi-independent of one another, a common factor analysis was performed on the items of the two scales: BDI-13 and SOC-13 (KMO = 0.91, Bartlett test: *p* < 0.001). Using a two-factor solution, the results of the principal component analysis showed that the first factor included all of the items listed on the BDI-13 scale, with failure, sadness, difficulty making decisions, and difficulty getting work done having the highest factor loadings. The second factor including all but two of the items on the SOC-13 list (not included: “until now your life has had very clear goals and purpose vs. no clear goals or purpose” and “doing the things you do every day is a source of deep pleasure and satisfaction vs. a source of pain and boredom”) was easy to isolate from items composing the tendency to depression (Table 5 and Figure 1).

Within component 2 (SOC), the items referring to comprehensibility had the highest factor loadings. The two factor values obtained (DEP-factor and SOC-factor_rev) were thus able to measure the tendency to depression and the sense of coherence nearly independently of one another. Inserting these results into the depression and sense of coherence variables of model 3 in Table 4, the results shown in Table 6 were obtained. 

According to these results (Table 6), both factors are significant, meaning that sense of coherence and tendency to depression, when treated independently, have a significant influence on the respondents’ odds of addiction. Both a weak sense of coherence and a tendency to depression increase the chances of addiction, while a strong sense of coherence diminishes the effects of depression.

For the motivations of consumption, a significant difference was found between daily users vs. those who consumed EDs less frequently. Controlling for sex and age, frequent users marked taste, stimulation, thirst, and work-out much more often than their counterparts did in the infrequent consumer group. Therefore, those who consumed EDs mostly for the reasons listed above were more likely to become addicted. 

Considering that enhanced performance and before working out were motivations commonly indicated by respondents, the effect of participation in sports was examined regarding the odds of addiction. The response to the question “Do you practice sports?” turned out to be a significant factor (p = 0.046) in the logistic regression model that featured sex, age, and other motivations together. According to this model, participation in sports reduced the chance of ED addiction by about fifty percent (AOR = 0.47, 95% CI: 0.22–0.80).

Interestingly, the rate of ED consumption among those who regularly participated in sports activities was high; approximately 29.3% (95% CI: 25.5–33.4 %), and 16.9 % (95% CI: 25.5–33.4 %) of them were almost daily users. 

## 4. Discussion

The rate of ED consumption has risen drastically throughout Europe, and these beverages are increasingly popular among young people. This fact is something all studies agree on [36,37]. In our sample of Hungarian high school and college students (aged 17–26 years), we found that rates of ED consumption matched the international data.

Roughly one fifth of the sample reported using EDs on a regular (almost daily) basis. 

Rates of ED consumption were highest in the sample of high school students, especially among male respondents. It is also well known that males are more likely to report ED use and are also more likely to admit higher levels of consumption than females [10,38,39].

Twenty-four percent (95% CI: 17.9-30.0 %) of consumers within our sample used EDs alongside, or mixed with alcohol. In another study, 23.2% of high school students (15–19 years) admitted consuming ED drinks with alcohol [27], and young regular consumers of EDs were also more likely to consume larger quantities of alcohol per occasion, and were also more prone to make poor decisions (e.g., drunk driving) or engage in aggressive behavior [11]. The same young people were also at increased risk for excessive alcohol consumption later in life [40].

According to several studies, [10,41], primary motivations for ED use are taste and energy/stimulation. Those participating in sports, especially males, used EDs to improve sports performance [10]. According to a study by [39], most users consumed EDs because of sleep deprivation (67%), to be energized (65%), and alongside alcoholic drinks at parties (54%). The most common motivations for ED use in the sample were also taste and fatigue. Young people who marked taste and thirst as their primary motivation for ED consumption were more likely to become addicted, because those who marked these as their primary motivations were more likely to consume ED on a daily basis than their counterparts whose motivations differed.

The social environment of respondents and the amount of time they spent with friends were also factors influencing ED use. Parents supporting or forbidding ED use have a significant influence on their children [42]. According to our data, parents, siblings, and friends all influenced decisions regarding ED consumption.

Among the health psychology factors tested, sense of coherence (independent of sex and age) of respondents had the strongest influence on their chances of becoming ED users, and sense of coherence showed a strong correlation with the appearance of depression symptoms. Those with a weaker sense of coherence and a tendency toward depression were much more likely to become addicted. Those who were active in sports were significantly less likely to report symptoms of depression, and their sense of coherence was also stronger than that of young people who did not engage in sporting activities. At the same time, ED consumption was very common among young people who were active in sports, so sport did not significantly protect young people from ED use.

Respondents, and not only daily users, reported experiencing several negative side effects following ED consumption, just as in other studies [41,42].

## 5. Conclusions

To protect young people, especially teenagers, from excessive ED consumption, parents and teachers should strive to improve the sense of coherence of children both at home and in school from a very early age, especially considering the difficulties teenagers and young adults have [43].

Parents and other adults should not support or enable children’s ED consumption. Recognizing the adverse health effects of EDs and preventing its usage is important, because those who have easy access to these products (i.e., their parents and siblings are also users and EDs are kept at home) are more likely to acquire the habit of frequent, even excessive consumption. It is especially important to alert people to the detrimental effects of simultaneous ED and alcohol consumption. 

## 6. Strengths and Limitations of the Study

To our knowledge, this is the first study examining the patterns of ED consumption and the odds of excessive consumption in connection to Antonovsky’s salutogenic sense of coherence. Since depression is a key factor in addiction formation, our study highlights the fact that a strong sense of coherence can counter some of the effects of depression. Furthermore, it is important to emphasize that health education and health promotion must strive, indirectly or directly, to increase young people’s sense of coherence. A limitation of the present study is its cross-sectional design, which hampers making causal inferences. 

## Figures and Tables

**Figure 1 ijerph-17-01290-f001:**
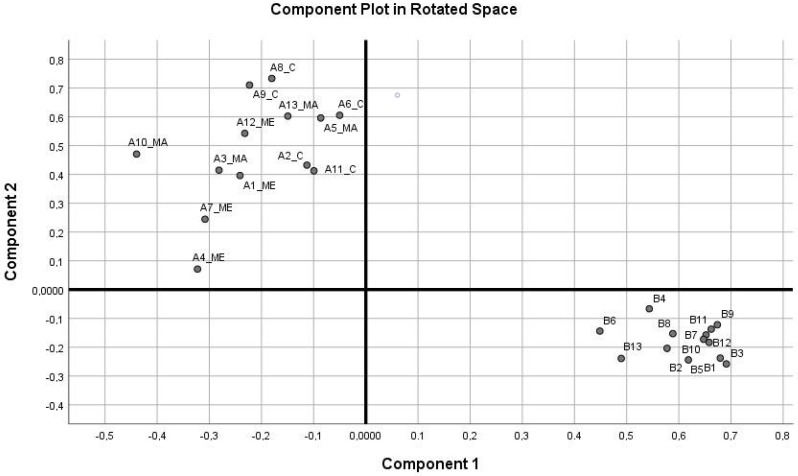
Results of the principal component analysis.

**Table 1 ijerph-17-01290-t001:** Characteristics of the sample according to sex.

Characteristics	Male (n = 284)		Female (n = 347)		Statistics	Total (n = 631)	
	Mean	SD	Mean	SD	*p* (U test)	Mean	SD
Age	19.5	2.0	19.2	2.0	n.s.	19.3	2.0
Depression	3.8	4.3	5.9	5.9	<0.001	4.9	5.4
Sense of coherence	59.0	10.3	55.2	11.5	<0.001	56.9	11.1
Comprehensibility	21.8	4.8	19.3	5.1	<0.001	20.5	5.1
Manageability	18.4	4.2	17.1	4.3	<0.001	17.7	4.3
Meaningfulness	18.7	4.2	18.8	4.2	n.s.	18.8	4.2
Energy drink use	n	%	n	%	*p* (Wald stat.)	n	%
Does not consume energy drinks	45	15.8	79	22.8	---	124	19.7
Very rare consumption	118	41.5	193	55.6	0.748	311	49.3
1-2 times per month	54	19.0	24	6.9	0.000	78	12.4
1-2 times per week	34	12.0	31	8.9	0.035	65	10.3
More than twice a week	8	2.8	5	1.4	0.085	13	2.1
Once a day	13	4.6	9	2.6	0.049	22	3.5
More than once a day	12	4.2	6	1.7	0.019	18	2.9
The following consume energy drinks regularly					*p* (chi2 test)		
Parents (yes-no)	57	20.1	49	14.1	0.047	106	16.8
Sibling (yes-no)	117	41.2	123	35.4	n.s.	240	38.0
Friend/boyfriend/girlfriend (yes-no)	149	52.5	220	63.4	0.006	369	58.5
Availability of energy drinks at home					*p* (chi2 test)		
Yes	87	30.6	78	22.5	n.s.	165	26.1
No	193	68.0	264	76.1		457	72.4

**Table 2 ijerph-17-01290-t002:** Motivations for energy drink consumption according to sex.

Reasons for Energy Drink Consumption	Males (n = 121)		Females (n = 75)		*p* (chi2 Test)
	%	95% CI	%	95% CI	
For fun	16.5	10.9–24.1	5.3	2.1–12.9	0.020
Tastes good	43.8	35.3–52.7	56.0	44.8–66.7	ns
Stimulation	16.5	10.9–24.1	18.7	11.5–28.9	ns
Fatigue	33.1	25.3–41.9	64.0	52.7–73.9	<0.010
Revs me up	17.4	11.6–25.1	21.3	13.6–31.9	ns
Quenches thirst	11.6	7.0–18.5	12.0	6.4–21.3	ns
Cool/trendy	2.5	2.4–7.0	0.0	0.0–4.9	ns
Enhanced performance	12.4	7.7–19.4	6.7	2.9–14.7	ns
Work-out	11.6	7.0–18.5	2.7	0.7–9.2	0.027

**Table 3 ijerph-17-01290-t003:** Short-term adverse effects of energy drink consumption.

Which Adverse Effects Did You Experience Following Energy Drink Consumption?	Males (n = 121)		Females (n = 75)		*p* (chi2 Test)
	%	95% CI	%	95% CI	
Headache	21.5	15.1–29.6	16.0	9.4–25.9	ns
Nausea	11.6	7.0–18.5	6.7	2.9–14.7	ns
Weakness	12.4	7.7–19.4	14.7	8.4–24.4	ns
Tremors	28.1	20.9–36.7	30.7	21.4–41.8	ns
Dizziness	8.3	4.6–14.6	13.3	7.4–22.8	ns
Loss of consciousness	2.5	0.9–7.0	0.0	0.0–4.9	ns
Insomnia	26.4	19.4–34.9	38.7	28.5–49.9	ns
Irritability	2.5	0.9–7.0	8.0	3.7–16.4	ns
Tachycardia	32.2	24.6–40.1	38.7	28.5–49.9	ns
Breathlessness	2.5	0.9–7.0	4.0	1.4–11.1	ns
Fear	0.8	0.2–4.5	8.0	3.7–16.4	0.013
Diarrhea	1.7	0.5–5.8	2.7	0.7–9.2	ns

**Table 4 ijerph-17-01290-t004:** The binary logistic regression analysis of depression and sense of coherence to energy drink addiction, controlled for sex and age.

**Model 1**	**B**	**S.E.**	**Sig.**	**Exp(B)**		
					**Lower**	**Upper**
Sex	−0.745	0.228	0.001	0.475	0.304	0.743
Age	−0.261	0.064	0.000	0.770	0.679	0.874
Depression	0.059	0.019	0.002	1.060	1.022	1.100
Model 2	B	S.E.	Sig.	Exp(B)	95% C.I. for EXP(B)	
					Lower	Upper
Sex	−0.778	0.229	0.001	0.460	0.293	0.720
Age	−0.242	0.065	0.000	0.785	0.692	0.891
Sense of coherence_rev *	0.037	1.402	0.000	1.037	1.017	1.058
Model 3	B	S.E.	Sig.	Exp(B)	95% C.I. for EXP(B)	
					Lower	Upper
Sex	−1.232	0.206	0.000	0.292	0.195	0.437
Age	−0.206	0.053	0.000	0.813	0.733	0.903
Depression	0.026	0.023	0.260	1.026	0.981	1.074
Sense of coherence_rev *	0.029	0.012	0.017	1.029	1.005	1.054

Dependent variable: Daily vs. less frequent energy drink consumption. * We revised the SOC scale in such a way that a higher score indicates a lower sense of coherence.

**Table 5 ijerph-17-01290-t005:** Rotated component matrix ^a^.

Item with Factor Loadings > 0.600	Component	
	1	2
B3 Past Failure	0.691	
B1 Sadness	0.679	
B9 Indecisiveness	0.674	
B11 Tiredness or Fatigue	0.662	
B12 Loss of Energy	0.658	
B7 Suicidal Thoughts or Wishes	0.652	
B10 Change in body image	0.647	
A8_C Do you have very mixed-up feelings and ideas?		0.733
A9_C Does it happen that you have feelings inside you would rather not feel?		0.710
A6_C Do you have the feeling that you’re being treated unfairly?		0.605
A13_ME How often do you have feeling that you’re not sure you can keep under control?		0.603

Extraction Method: Principal Component Analysis. ^a^ Rotation converged in 3 iterations.

**Table 6 ijerph-17-01290-t006:** Binary logistic regression analysis of the depression-factor and sense of coherence-factor to energy drink addiction, controlled for age and sex.

Variables	B	S.E.	Sig.	Exp(B)	95% C.I. for Exp(B)	
					Lower	Upper
Sex	−0.906	0.226	0.000	0.404	0.259	0.629
Age	−0.270	0.065	0.000	0.763	0.672	0.866
Dep-factor	0.251	0.099	0.011	1.285	1.058	1.561
SOC-factor_rev *	0.333	0.109	0.002	1.395	1.127	1.726

Dependent variable: Daily vs. less frequent energy. * A lower sense of coherence is matched with a higher score.

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
