# Peer review of "Energy Drink Consumption, Depression, and Salutogenic Sense of Coherence Among Adolescents and Young Adults"

_ijerph, 2020, doi:10.3390/ijerph17041290_

Round 1
Reviewer 1 Report
The data presented in this study are well analyzed. In this manuscript, the authors reported that motivations and choices of energy drink consumption by young populations from Hungarian; they have excessive consumption of energy drink, particularly among males, with negative health consequences.
Concerns:
1.- the section on "DISCUSSION" is not relevant statements
2.- In table 1, the rate of depression in males, please use 3.8 instead of 3,8. same errors are also seen in tables 2,3

Reviewer 2 Report
Please find comments in attached document.

Round 2
Reviewer 2 Report
The authors have addressed my concerns.